# Dopaminergic Epistases in Schizophrenia

**DOI:** 10.3390/brainsci14111089

**Published:** 2024-10-29

**Authors:** Adela Bosun, Raluka Albu-Kalinovic, Oana Neda-Stepan, Ileana Bosun, Simona Sorina Farcas, Virgil-Radu Enatescu, Nicoleta Ioana Andreescu

**Affiliations:** 1Doctoral School, “Victor Babes” University of Medicine and Pharmacy, 300041 Timisoara, Romania; adela.bosun@umft.ro (A.B.); raluka.kalinovic@hosptm.ro (R.A.-K.); oana.neda-stepan@umft.ro (O.N.-S.); 2Eduard Pamfil Psychiatric Clinic, Timisoara County Emergency Clinical Hospital, 300425 Timisoara, Romania; enatescu.virgil@umft.ro; 3Department of Neurosciences, Discipline of Psychiatry, “Victor Babes” University of Medicine and Pharmacy, Eftimie Murgu Square 2, 300041 Timisoara, Romania; 4Department of Ophthalmology, Clinical Hospital “Cai Ferate”, 300173 Timisoara, Romania; ileanabosun@yahoo.com; 5Department of Microscopic Morphology, Discipline of Genetics, Genomic Medicine Centre, “Victor Babes” University of Medicine and Pharmacy, 2 Eftimie Murgu Square, 300041 Timisoara, Romania; andreescu.nicoleta@umft.ro; 6Regional Center of Medical Genetics Timis, Clinical Emergency Hospital for Children “Louis Turcanu”, Iosif Nemoianu Street N°2, 300011 Timisoara, Romania

**Keywords:** dopaminergic theory, schizophrenia genetics, epistasis, dopamine imbalance, psychiatry genetics

## Abstract

**Background:** The dopaminergic theory, the oldest and most comprehensively analyzed neurotransmitter theory of schizophrenia, remains a focal point of research. **Methods:** This systematic review examines the association between combinations of 14 dopaminergic genes and the risk of schizophrenia. The selected genes include dopamine receptors (DRD1–5), metabolizing enzymes (COMT, MAOA, MAOB, DBH), synthesizing enzymes (TH, DDC), and dopamine transporters (DAT, VMAT1, and VMAT2). **Results:** Recurring functional patterns show combinations with either hyperdopaminergic effects in limbic and striatal regions or high striatal and low prefrontal dopamine levels. The protective statuses of certain alleles or genotypes are often maintained in epistatic effects; however, exceptions exist. This complexity could explain the inconsistent results in previous genetic studies. Investigating individual alleles may be insufficient due to the heterozygous advantage observed in some studies. **Conclusions:** Schizophrenia may not be a monolithic disease, but rather a sum of different phenotypes which respond uniquely to different treatment and prevention approaches.

## 1. Introduction

Schizophrenia (SZ) is a chronic and often debilitating disease with a complex etiopathogenesis that involves genetic, neurochemical, and environmental factors [1]. There is considerable interest in establishing a genetic basis for the disease, but the results are still far from being conclusive, probably due to the large impact of environmental factors, including childhood trauma [2] and stress [3].

The oldest and most comprehensively analyzed neurotransmitter theory of schizophrenia is the dopaminergic theory [4], which posits that imbalances of this compound are at the core of the disorder. It is hypothesized that a heightened sensitivity of receptors to dopamine at the subcortical level would give rise to positive symptoms, such as hallucinations or delusions, while lower prefrontal dopamine levels are responsible for negative (lack of motivation, blunted affect, etc.) and cognitive deficits [5].

Epistasis is the phenomenon in which the effect of one gene is altered or masked by one or more other genes, influencing the overall expression of a trait, or, in a more general manner of speaking, the interaction between two or more genes to influence a trait. As the dopaminergic system is thought to play a key role in the development of schizophrenia and psychotic disorders yet no single gene has been discovered to confer a clear vulnerability to the disorder [6], we decided to investigate from a gene–gene interaction perspective, since singular genes may be responsible for smaller effects, while a combined approach might reveal stronger effects and shed more light on the relationship between the dopaminergic system and schizophrenia.

The four major dopaminergic circuits in the brain are the mesolimbic, mesocortical, tubero-infundibular, and nigrostriatal pathways. The mesolimbic system is heavily involved in emotion processing [7] and is incriminated in the positive symptomatology in schizophrenia through over-reactivity to stimuli [8], while the mesocortical pathway is responsible for learning and reasoning [9] and is involved in the negative and cognitive symptomatology.

To the best of our knowledge, no other reviews have specifically examined the interactions between dopamine-related genes and schizophrenia, nor are there any broader reviews that address this particular research question.

### 1.1. Dopamine Receptors

Dopamine receptors are generally classified as D1-like (D1, D5) or D2-like (D2–D4). D1-like receptors stimulate adenylate cyclase via G_s_ and mediate excitatory neurotransmission. D2-like receptors inhibit cAMP formation and interact with other types of G proteins, namely G_i_ and G_o_ [10], and often function as autoreceptors. It is believed that prefrontal function is coregulated through the opposing D1- and D2-mediated actions [11].

DRD1, localized on chromosome 5, encodes the most abundant receptor in the brain [12,13,14] (including in the prefrontal cortex), which is predominantly an excitatory receptor. This, the D1 receptor, plays a crucial role in cognitive processes such as attention, executive function, learning [15], and working memory [16]. Among the gene variants potentially involved in developing schizophrenia, rs4532 and rs5326 [17] have been heavily investigated. A recent meta-analysis incriminates the GA genotype of rs5326, while rs4532 correlation did not survive the analysis [17]. The GG, AA, and T+ genotypes of rs1799914 [18], rs686 (in males), and rs10063995 (in females) [19], respectively, also seem to be associated with a higher incidence of the disease. Rs4532 is associated with antipsychotic response [17], tardive dyskinesia, [20] and (as with rs686) the quantity of tobacco used by SZ patients [21].

DRD2, which maps to chromosome 11 [22] and encodes dopamine receptor D2, is an autoreceptor that downregulates the synthesis and release of dopamine [23]. Its functions include modulating learning and memory, attention, and sleep [23], and it represents the main target of most antipsychotics. It has been found to be upregulated in the brains of patients with schizophrenia. It is unclear whether that is attributable to disease or the action of antipsychotics, as most studies have been conducted on patients that have been medicated throughout their life, and this finding is not consistent in drug-naive patients [24]. Rs2514218 located near the DRD2 gene is the only locus related to dopamine signaling that has been found by the Schizophrenia Psychiatric GWAS Consortium (PGC) to be a predictor of disease risk [25]. Among its polymorphisms, the *Ser*/*Cys311* variant is associated with disorganization and delusional ideation [26,27], while both *Ser*/*Cys311* and *Taq1A* (rs1800497) correlate with risk of schizophrenia [28,29]. As such, it may appear that variations in the DRD2 gene may confer a risk factor for developing schizophrenia, as well as having a modifier effect.

DRD3 is a D2-like receptor encoding gene located on chromosome 3 with a very high dopamine affinity (20 times higher than D2) [10]. D3 can function as an autoreceptor [30] and regulate the dopamine transporter [31] DAT (SLC6A3). Most studies have focused on rs6280 (*Ser9Gly*) with inconsistent results [32]. *Gly* homozygotes have a higher affinity for dopamine binding [33], and seem to have poorer executive functions, findings present both in healthy and psychosis-affected individuals [34].

DRD4, situated on chromosome 11, is involved in novelty seeking and addiction [35,36] and may play a role in developing the SZ, considering its high density in the frontal cortex and amygdala [37]. Human receptor D4 has a polymorphic region consisting of a variable number of repeats in exon 3 (D4D3) [10]. Results are inconsistent, but a few studies report that a higher number of repetitions is associated with risk of developing SZ, including with risk of delusions in affective psychoses [35,38,39].

The DRD5 gene encodes human dopamine receptor D5, which is involved in attention, working memory, and locomotion [15,40]. This receptor is not as well studied in SZ. A few studies have found associations between schizophrenia and rs1850744 [41], rs77434921, rs1800762 [42] and 148 bp-VNTR [43].

### 1.2. Metabolizing Enzymes Genes

In the human brain, dopamine is mainly metabolized by COMT, MAOs, and DBH. MAOs deaminate dopamine, producing DOPAL (an aldehyde), further converted to homovanillic acid by COMT and eliminated from circulation [44]. COMT is responsible for dopamine degradation in the prefrontal cortex by converting it to 3-methocxytyramine (generally considered physiologically inert) [45]. DBH (dopamine beta-hydroxylase) converts dopamine to noradrenaline [44].

Catechol-O-methyltransferase, encoded by the COMT gene, is an enzyme that inactivates dopamine (plus other catecholamines) in the prefrontal cortex, eliminating it from the system [46]. This enzyme exhibits sexual dimorphism, its functionality being inhibited by estrogen [47]; thus, we will emphasize the need to investigate its behavior in a sex-dependent manner. The most well-studied polymorphism is *Val158Met*, where *Val*, the ancestral variant [48], is associated with an almost 40% higher level of the enzyme, and better thermal stability (at 37 °C) [37], which results in faster dopamine metabolism in the prefrontal cortex. Data suggests an association between the *Val* allele and a higher risk of schizophrenia [49]. Some studies have shown an association between this COMT variant and negative symptomatology [49]. Particularly in females, the presence of the *Val* allele influences the severity of negative symptoms [50].

Monoamine oxidase A is an enzyme encoded by the MAOA gene that metabolizes dopamine, norepinephrine, and serotonin [51]. Low-functioning genetic variants are heavily associated with aggression [52] and, in conjunction with childhood trauma, antisocial behavior [51]. While this gene is not particularly associated with SZ, the rs6323 variant and a variable number of tandem repeats located near the promoter (uVNTR) have shown inconsistent association with disease risk, and rs6323 (G allele) may be responsible for affective symptomatology in patients [53,54,55]. Since both MAOA and MAOB are located on the X chromosome, genetic analyses should ideally be stratified by sex.

The MAOB gene encodes monoamine oxidase B, an enzyme responsible for the degradation of dopamine and associated with neurodegenerative diseases such as Alzheimer’s and Parkinson’s disease [56,57]. The data are scarce, but rs1799837 (G allele) [58] is suggested to be associated with SZ risk, rs1799836 (A allele) with alogia in male patients, and rs6651806 (A allele) with higher severity of negative symptoms (also in males) [50]. There are also some weak associations with delusions in patients [59].

Dopamine beta-hydroxylase, encoded by the DBH gene, converts dopamine to noradrenaline and is involved in addiction and the decision-making process [60,61]. The *5’ Ins*/*Del* polymorphism may be correlated with the severity of positive symptomatology [62] without directly influencing disease risk [63]. Along with rs6271 and rs1108580, it also appears to be somewhat associated with cognition in patients [64,65,66].

### 1.3. Synthesizing Enzymes

Dopamine is endogenously synthesized in the human brain from tyrosine, converted (through TH, tyrosine hydroxylase) to L-DOPA, and further converted to dopamine by DDC (DOPA decarboxylase) [44].

The TH gene encodes tyrosine hydroxylase, an enzyme that facilitates the conversion of tyrosine to L-DOPA [44] and is effectively a rate-limiting enzyme in dopamine synthesis [67]. A high-functioning VNTR variant in intron 1 appears to be associated with higher positive symptom dimension scores on the PANSS scale [67], while the CC (low-functioning) genotype of rs10770141 is associated with lower dopamine availability and lower IQ among SZ patients [68]. Some minor polymorphisms may also increase the risk of suicide attempts in patients [69]. TH is generally not considered to be associated with disease risk [70], but it may influence different dimensions of SZ symptomatology.

The DDC gene encodes dopa-decarboxylase, an enzyme which finalizes the dopamine synthesis process through the decarboxylation of L-DOPA to dopamine, as well as the serotonin synthesis process [71]. The T allele of rs2237457 is associated with lower gene expression and potentially treatment-resistant schizophrenia (TRS) [71]. The minor allele G of rs10499696 appears protective for TRS [72]. Two deletions, a 4 bp deletion in exon 1 and a 1 bp deletion in the promoter region, have been associated with earlier and later onset, respectively, of SZ in males [73].

### 1.4. Dopamine Transporters

Dopamine is reuptaken from the synaptic cleft through the action of the dopamine transporter, which delivers it back to the cytosol of the presynaptic neuron and is further packaged by VMAT to the synaptic vesicles, where it can be stored until further release [74,75].

The dopamine transporter (DAT) encoded by the SLC6A3 gene is responsible for synaptic dopamine reuptake. This gene heavily influences the intracellular dopamine levels in the brain as well as the intensity and duration of dopamine action [76]. A recent meta-analysis incriminated the A allele of rs2975226 and TT genotypes of rs464049 and rs3756450 to be potential risk factors for schizophrenia [77].

The SLC18A1 gene encodes the vesicular monoamine transporter 1 (VMAT1), primarily localized in neuroendocrine cells (such as the adrenal medulla), and is responsible for the transport of several monoamines into synaptic vesicles (dopamine, serotonin, adrenaline, noradrenaline, and histamine) [74]. It appears to play a role in frontal dopamine regulation, but the mechanisms are yet unclear [78]. *Thr4Pro* [79] and *Thr98Ser* (in females) [74] have been shown to be associated with schizophrenia.

VMAT2, encoded by the SLC18A2 gene, is involved in repackaging intracytoplasmic dopamine (as well as serotonin, dopamine, and norepinephrine) into vesicles for storage. Literature is scarce, but the AA genotype of rs363371 [80] seems to be protective in males, and rs363285 may influence executive function in patients [65].

Schizophrenia is a highly multifactorial disorder, and while numerous studies have focused on identifying genetic risk factors through single-gene analyses, this approach is overly simplistic and has often produced conflicting results, suggesting that it is unlikely to fully capture the complexity of the disease. However, a genetic basis for schizophrenia is well-established, as shown by evidence from twin and adoption studies [81], indicating that genetic vulnerability is likely polygenic [82].

Given the long-recognized association between schizophrenia and dopamine imbalances, we believe that exploring multigenic interactions among genes directly related to key components of the dopamine system (such as receptors, enzymes, and transporters) provides a valuable starting point for understanding the genetic basis of the disorder. This focus does not exclude the potential role of other genes or environmental factors, which should be further explored.

Research has shown that dopamine-related genes interact in ways that influence cognitive and behavioral processes relevant to schizophrenia. For example, interactions between DRD2 and COMT affect intra-network connectivity in the default-mode and salience networks [83], while DBH and MAOA influence attentional bias toward negative expressions [84]. Social cooperation has also been linked to interactions between DRD2 and COMT [85]. Additionally, DRD4 and COMT together impact prefrontal response control, even though each gene may not have a significant effect individually [86], and variations in DRD2 and COMT combinations are associated with differences in working memory performance [87].

There is also evidence of interactions among DRD2, DRD4, and COMT in the response to antipsychotic treatment [88]. For instance, the combination of the Met allele of COMT and the 120-bp allele of DRD4 is associated with better responses to clozapine, but only when these alleles are present together [89]. Therefore, investigating multigenic interactions is essential not only for identifying risk factors for schizophrenia but also for gaining a deeper understanding of the disease. In the context of dopamine-related gene interactions, such research could provide insights into the dopamine imbalances that underlie schizophrenia.

This systematic review investigated the association between combinations of 14 dopaminergic genes and the risk of schizophrenia. We chose the dopamine DRD1–5 receptors, dopamine metabolizing enzymes (COMT, MAOA, MAOB, and DBH), synthesizing enzymes (TH and DDC), and dopamine transporters (DAT, VMAT1, and VMAT2). While only some of these genes have shown promising results, we decided to include all of them considering the possibility of uncovering interactive effects. Since schizophrenia is a highly heterogenous disease, we are also interested in potentially identifying combinations related to specific phenotypes.

## 2. Methods

### 2.1. Selection Criteria

As per Huge Reviews guidelines [90], we attempted to formulate criteria to be as inclusive and sensitive as possible because resources on the subject are scarce. Since SZ is heavily influenced by environmental factors, we would like to include its potentially latent forms, and we have decided to include intermediate phenotypes for schizophrenia, as well as psychotic symptoms in other diseases, such as bipolar [91] or major depressive disorder. Conversely, dementia with psychotic symptoms is negatively correlated with a genetic risk for schizophrenia [92]; therefore, we will exclude these results. We have decided to exclude GWAS, as the results are generally hard to interpret regarding genetic interaction.

Therefore, we surveyed available literature for clinical studies on human populations that could link the presence of specific combinations of the genes of interest to either risk of developing schizophrenia or a protective effect against the disease. Eligibility criteria for inclusion were (1) case–control or correlational studies that evaluated the interaction between at least 2 of the 14 investigated genes; (2) all variants of the 14 investigated genes; (3) evaluating risk of schizophrenia, psychosis, psychotic symptoms, schizotypy, or intermediary phenotypes; (4) any language; (5) any age; (6) both sexes; and (7) any race.

The exclusion criteria were (1) non-human subjects (animal models), (2) reviews and meta-analysis, (3) conference abstracts, and (4) GWAS.

### 2.2. Identifying Studies

This systematic review adheres to the PRISMA guidelines: http://www.prisma-statement.org (accessed on 12 March 2023). A total of 273 individual searches have been performed until September 2023 in PubMed (5 September), Scopus (12 September), and Web of Science (5 September). The searches were composed of complex keywords for:1.each combination of 2 of the 14 genes under review (totaling 91 different complex keywords) connected through the operator “AND”; the complex keywords associated with each gene are as follows:(DRD1 OR “dopamine receptor D1” OR “dopamine D1 receptor”)(DRD2 OR “dopamine receptor D2” OR “dopamine D2 receptor”)(DRD3 OR “dopamine receptor D3” OR “dopamine D3 receptor”)(DRD4 OR “dopamine receptor D4” OR “dopamine D4 receptor”)(DRD5 OR “dopamine receptor D5” OR “dopamine D5 receptor”)(COMT OR Catechol-O-Methyltransferase)(“MAO-A” OR “monoamine oxidase A” OR “MAO A” OR MAOA)(“MAO-B” OR “monoamine oxidase B” OR “MAO B” OR MAOB)(DBH OR “Dopamine beta-hydroxylase” OR “dopamine beta-monooxygenase”)(TH OR “tyrosine hydroxylase” OR “tyrosine 3-monooxigenase”)(DDC OR “dopa decarboxylase” OR “tryptophan decarboxylase” OR “Aromatic L-amino acid decarboxylase” OR AADC OR AAAD OR “5-hydroxytryptophan decarboxylase”)(DAT OR “dopamine transporter” OR “dopamine active transporter” OR SLC6A3 OR “Solute Carrier Family 6 Member 3” OR “DA Transporter”)(VMAT1 OR “Vesicular monoamine transporter 1” OR “chromaffin granule amine transporter” OR CGAT OR “solute carrier family 18 member 1” OR SLC18A1 OR “VMAT 1” OR “VMAT-1”)(VMAT2 OR “Vesicular monoamine transporter 2” OR “solute carrier family 18 member 2” OR SLC18A2 OR “VMAT 2” OR “VMAT-2”)2.psychosis: “schizo* OR psychosis OR psychotic”,3.risk: “vulnerability OR predisposition OR susceptibility OR risk OR proneness”, and4.epistatic interaction: “epistasis OR “genexgene” OR “gene-gene” OR interact* OR epistatic OR GxG”,

Connected with the operator “AND”, without use of database filters, ex., (DRD1 OR “dopamine receptor D1” OR “dopamine D1 receptor”) AND (DRD2 OR “dopamine receptor D2” OR “dopamine D2 receptor”) AND (schizo* OR psychosis OR psychotic AND (vulnerability OR predisposition OR susceptibility OR risk OR proneness) AND (epistasis OR “genexgene” OR “gene-gene” OR interact* OR epistatic OR GxG). A manual search has also been performed. A total of 881 articles were identified, consisting of 227 individual studies after removing duplicates. After reviewing the titles and abstracts of those, we excluded 186 articles that did not mention genetic testing for assessing risk of any psychosis-related issues. The remaining 41 articles were read in their entirety, and a total of 18 articles were selected (Figure 1).

### 2.3. Data Collection and Analysis

Studies that were retrieved from database searching were collected in Zotero and then exported in a data collection form that was initially piloted for a smaller number of studies. Studies were abstracted by title, authors, year of publication, number of participants by group and sex, average age, diagnosis, ethnicity, investigated genes and polymorphisms, method, study design, and results. Regarding results, we abstracted the following information: genes and specific variants involved, degree of correlation (we have included trend-level associations), or the lack of association.

For the risk of bias assessment in case–control studies, we used the Newcastle–Ottawa scale (NOS) [93], which assesses quality based on three main categories: selection of study groups, comparability of groups, and ascertainment of exposures. Studies deemed “good quality” generally had robust sampling methods, clear criteria for genetic testing, and appropriate adjustment for confounding variables. Studies rated as “good quality” typically featured well-defined inclusion criteria and adequate control for confounders (such as age, gender, and socioeconomic status). Most studies were of good quality (13) with good selection, comparability, and exposure overall; 3 were identified as high-risk of bias (poor comparability [94,95,96], with one study having relatively weak selection criteria [96]); none were considered subject to a very high risk of bias. One correlational study [97] could not be evaluated using the NOS scale, and its main limitation was the lack of a population representative of the general public, as the sample primarily consisted of academic staff and students, particularly from psychology. Additionally, schizotypy was assessed through a self-report scale, which is less reliable than DSM-based diagnoses. One study had a small sample size [94], which may lead to less reliable conclusions due to low statistical power, increasing the risk of false positives or false negatives. Additionally, most studies focused on a single population, meaning that the findings might only be applicable to those specific groups, as different genetic backgrounds and environments could influence gene interactions. Due to the genetic genotyping methods used, all studies received the maximum score for exposure. It is also important to consider that studies with positive results are more likely to be published, which could lead to an overestimation of the effects reported in this review.

## 3. Results

### 3.1. Characteristics of Studies

The 18 studies selected (Table 1) were published between 1996 and 2020 and analyzed a total of 13,025 participants: 5066 patients, age 39.9 (8.3), and 5027 controls, age 40.8 (8.7). The analyses were conducted on Caucasian, Asian, and Indian populations, with a slight prevalence of male participants (52.09%). The main diagnosis for case groups was schizophrenia (according to DSM-IV criteria).

A total of 106 variants in 12 genes have been studied (Table 2), and the most-used methods were logistic regression, multifactor dimensionality reduction (MDR), and Chi-square test. Very few studies conducted separate analyses stratified by sex.

Out of the 14 genes surveyed, significant epistatic interactions frequently implicated COMT, SLC6A3 (DAT1), DRD2, and DRD3 (Figure 2); furthermore, rs4680, rs6347, rs1800497, and rs6280 were involved in more than one interaction. While the total number of possible combinations would be 91, only 15 appear potentially of interest, out of which only COMT–DRD2, COMT–DAT, COMT–TH, and DRD3–DAT have significant interactions in more than one study (Figure 3).

### 3.2. Synthesis–Receptor–Transporter–Metabolism Gene Complex Interaction

An interaction cluster mainly composed of dopaminergic elements appears to predispose patients to a schizophrenia phenotype characterized by the predominance of delusional ideation [104]. Three main phenotypical clusters were identified in an Australian population, mainly characterized as follows: speech disorder and affective symptoms cluster, hallucinations cluster, and delusions cluster.

The speech disorder/affective cluster includes speech incoherence, thought disorder (thought echoing and thought withdrawal), agitation, blunted affect, late insomnia, nihilistic delusions, and delusions of poverty. The second cluster mainly consists of hallucinations (abusive and commentative), as well as perceptual and bizarre delusions, thought disorder (insertion and broadcast), modified appetite, excessive sleep, and initial and middle insomnia.

The third cluster is predominantly defined by delusions and is represented by symptoms such as delusions of influence, grandiose and persecutory delusions, suicidal ideation, and also dysphoria, anhedonia, lack of energy, and poor concentration. While delusions are mainly classified in the third cluster, nihilistic/poverty and bizarre/perceptual delusions have been integrated into the first and second cluster, respectively.

A linear regression model identified an SNP combination pattern that can be linked to the presence of delusional symptomatology, mostly composed of genes involved in the dopaminergic system (in order of influence): COMT gene rs4680, SLC6A3 (DAT1) gene rs6347, DDC gene rs1966839, and DRD2 gene rs2734839 (Figure 4).

Other genes identified as part of this complex are the GRM3 gene, NOS1AP gene, and HTR2A gene, implying the minor involvement of glutamate, nitric oxide, and serotonin in this complex interaction. This study suggests that dopaminergic dysfunction is related to a specific phenotype of schizophrenia, one majorly defined by the presence of delusions. Gender-specific effects are not accounted for, which is unfortunate given the involvement of the COMT gene in the result, especially since the cluster of interest is mainly composed of males (71%).

Also, it is of note that the genetic combination pattern solely associated with SZ disease risk is composed of 10 SNPs, out of which only 4 are directly related to the dopaminergic system: DDC (rs2329371, rs1966839), SLC6A3 (DAT1) (rs11133767), and COMT (rs4646316), hinting at the probable multi-neurotransmitter imbalance pathogenesis of schizophrenia.

### 3.3. Receptor-Metabolism Gene–Gene Interaction

The COMT *Val158Met* polymorphism stands out, being involved in almost all identified gene–gene interactions, and two of the studies included incriminated the combination of DRD2 *Taq1A* (rs1800497) and COMT *Val158Met* (rs4680), functionally translated by low DRD2 density and rapid prefrontal dopamine degradation.

The *Val*/*Val* genotype of *Val158Met* in carriers of the A1 variant of *Taq1A* was associated with double (OR 2.01) SZ risk in a Japanese population [102]. The functional behavior of the *Val*/*Val* genotype is well known, and this allele results in an important enzymatic activity increase in the COMT enzyme, the main dopamine metabolizer in the prefrontal cortex. It is suspected that this may lead to increased dopamine release in the nucleus accumbens. Individuals with the minor A1 allele of the *Taq1A* polymorphism have 12% less striatal DRD2 availability, potentially linked to increased dopamine synthesis due to decreased presynaptic autoreceptor function (Figure 5). While this combination is almost twice as common in patients than controls, it only accounts for 30% of SZ patients, supporting the involvement of multiple genetic patterns and environmental influence in disease development. No associations have been found between this combination and the different clinical subtypes of SZ (paranoid, disorganized, etc.). Gender-specific analysis has not been performed.

Interestingly, an MDR analysis in a Russian population in Central Asia incriminated the opposite genotypes of *Taq1A* and *Val158Met*, specifically *A2*/*A2* and *Met*/*Met* [111]. The functional result of the *A2*/*A2* and *Met*/*Met* combination is higher dopamine autoreceptor function and impaired dopamine degradation (Figure 5). With an OR of 3.25, this finding implies that the genetic conformation of Russians confers sensitivity to slower dopamine turnover and higher autoreceptor density. The *A1* allele and *Val*/*Val* genotype are protective in this population as *A2*/*A2*, and *Val*/*Met* also has a rather high OR of 2.27. These results are not replicated in the Tatar group [111], possibly due to different environmental and cultural factors or competing genetic interactions.

No epistatic effects between Taq1A and Val158Met have been observed in Indian [95] and Australian [104] studies that have investigated multiple interactions. Taq1A Val/Met interactions may not have been investigated in Indian populations because the criteria for selecting interactions were not clearly stated; thus, it is possible that only SNPs significant for single-gene testing were included for further interaction analysis. In Australian populations, the large number of genes analyzed under linear regression may have led to the exclusion of Taq1A Val/Met interactions, as any potential associations could have been overshadowed by larger combinations of genes (5 SNPs or more).

Another DRD2 and COMT gene–gene interaction was identified by an MDR analysis in a Southern Indian group [103]. Out of 16 analyzed polymorphisms of DRD2, COMT, and BDNF, the model selected DRD2 rs6275 × COMT rs4680 as the best. Rs4680 (GG) and rs6275 (TT) are incriminated as risk genotypes, while heterozygotes have relatively lower risks. It is of note, however, that the highest risk combination (OR 4.5) consists of risk rs6275 TT and protective genotype rs4680 AA. A more complex model identified (albeit only of trend significance) is DRD2 rs6275 × DRD2 rs4274224 × COMT rs4633 × COMT rs3788319. No interactive effects of rs6275 × rs4680 have been observed in a Russian/Tatar population [111].

COMT rs4680 polymorphism also appears to interact with DRD2 rs2283265 in influencing the risk of hypofrontality (an endophenotype often associated with schizophrenia), as observed in an Australian study [94]. A higher load of risk alleles (rs4680 G, and rs2283265 T) is associated with poorer prefrontal activation in healthy controls, as measured on fMRI on prefrontal activation tasks. This effect is not apparent in medicated patients, which have scored much lower regardless of allelic load. It is unclear if this interaction is only associated with hypofrontality in healthy subjects, or if the variability can be negated by treatment.

An association study on Spanish participants uncovered a potential epistatic interaction between DRD1 (rs11746641 and rs11749676) and COMT (rs4680); interestingly, this was only in the male cohort [110]. It appears that in the case of *Val* homozygotes (of rs4680), the presence of the G allele of rs11746641 is associated with an increased risk of developing schizophrenia. A similar effect is also valid in the case of rs11749676, as the GG genotype appears to interact with (rs4680) *Val* homozygosity, leading to a higher SZ risk. The functionality of rs11749676 may be related to its strong linkage disequilibrium with other DRD1 variants that have been heavily investigated in psychiatric genetic studies. For example, the rs11749676 protective minor allele (A) is linked to the rs686 allele associated with lower DRD1 expression [110]. Therefore, it is possible that, in males, the combination of higher DRD1 expression and higher dopamine turnover in the prefrontal cortex may be a risk factor for schizophrenia.

### 3.4. Receptor–Transporter Gene–Gene Interaction

A Spanish team has explored the interactive effects of DRD3 *Ser9Gly* (rs6280) and SLC6A3 (DAT1) *uVNTR*, revealing no direct independent association with SZ of either but a significant interactive effect [101]. Specifically, the presence of a nine-repeat SLC6A3 (DAT1) gene variant (associated with lower DA transporter expression), either strongly protects or increases the risk of SZ depending on the DRD3 variant (*Ser*/*Ser*, and *Gly*/*Gly*, respectively). In this case, the risk combination would be defined by higher DA availability combined with the lower DA affinity of Gly-carriers.

DRD3 *Ser9Gly* also appears to interact with SLC6A3 (DAT1) rs12516948 in influencing SZ risk in Americans and Bulgarians; another combination found is of the less-investigated variants DRD3 rs10934256 × SLC6A3 (DAT1) rs463379 [98]. DRD3 rs1800828 displays interactive effects with VMAT2 rs363227 and (on a trend-level) with VMAT2 rs929493 in American and Bulgarian populations [98].

### 3.5. Receptor-Synthesis Gene–Gene Interaction

The presence of both the A1 allele of the DRD2 *Taq1A* polymorphism and T allele of the TH gene rs10770141 (*C*-*824T*) was associated with almost double the risk for SZ in a Japanese analysis [102]. The functional result of this combination would be higher DA synthesis, contrasted by lower receptor binding.

### 3.6. Transporter-Metabolism Gene–Gene Interaction

The less-studied COMT rs174696 and SLC6A3 (DAT1) rs464049 appear to interactively influence the risk of schizophrenia, but their functionality is yet unknown [98]. Other trend-level interactions in both American and Bulgarian groups worth mentioning are between COMT rs174696 and two other SLC6A3 (DAT1) variants (rs463379 and rs456082), as well as SLC6A3 (DAT1) rs464049 × COMT rs165815.

There may be some synergistic interaction between the three-repeat variant of MAOA uVNTR and SLC6A3 (DAT1) 67A/T, but the results are not statistically significant [105].

### 3.7. Synthesis-Metabolism Gene–Gene Interaction

An Indian team investigated 31 SNPs of nine DA genes, out of which only one potential epistasis was identified, namely between TH rs6356 and COMT rs362204 [95]. This result may indicate another interaction related to the overall DA quantity, as TH is effectively a rate-limiting enzyme, while COMT metabolizes and eliminates dopamine. However, while the GG variant of rs6356 appears protective, the enzyme variants do not have any major differences, suggesting a potential linkage disequilibrium phenomenon. The functionality of COMT rs362204 is currently unknown.

There is a trend-level interaction between TH gene rs10770141 (*C*-*824T*) and COMT rs4680 (*Val158Met*) in a Japanese population, with the T+ allele and *Val*/*Val* genotype conferring moderate risk (OR 1.43) [102].

### 3.8. Transporters Gene–Gene Interaction

A potential epistasis of the dopamine transporter and vesicular transporter VMAT2 may have been uncovered in an American/Bulgarian sample [98]. Specifically, rs6347 (DAT1) may interact with rs363338 (SLC18A2) in influencing SZ risk.

### 3.9. Receptor Gene–Gene Interaction

A potential interaction between dopamine receptors has been observed in a Korean population, more specifically between DRD3 Ser9Gly, DRD4 12 bp-repeat, and HTR2A Ser34 *=* (2A serotonin receptor gene), suggesting a larger dopamine–serotonin pattern [99]. No association was found between DRD3 Ser9Gly and the DRD4 12 bp-repeat when analyzed individually in a Japanese cohort [106]. This absence of an association might be due to differences between Korean and Japanese populations; however, it is also possible that the interaction between DRD3 and DRD4 in relation to schizophrenia risk is influenced by the serotonergic system, making it less significant when considered in isolation.

### 3.10. Metabolism Gene–Gene Interaction

The interaction of MAOA and COMT genes may follow an interesting pattern where concurrent effects (either high or low metabolizers of both) may lead to higher levels of schizotypy (as measured by the O-LIFE questionnaire in a German population) [97]. Individually, *Val*/*Val* homozygotes appear to have higher scores for unusual experiences (while heterozygotes overall have the lowest scores), and the low-functioning variant of the MAOA uVNTR is correlated in males with cognitive disorganization and introversive anhedonia. However, while not statistically significant, the interactive model posits that the concomitant presence of both risk genotypes is potentially somewhat protective, likely due to their opposite effects on dopamine levels. This effect is also present in the *Met*/*Met* and high-functioning uVNTR sample. Conversely, the combination of *Val*/*Val* and the high-activity uVNTR variant functionally leads to overall higher dopamine turnover and the potential build-up of neurotoxic metabolites, while *Met*/*Met* and low-activity uVNTR may result in higher dopamine availability due to blunted metabolism. These two “extreme” conformations appear to have higher scores in the “Unusual Experiences” subscale.

While the interaction between COMT Val158Met and MAOA uVNTR may be linked to schizotypy, this association has not been found to correlate with the development of schizophrenia in Indian [95] and English [109] populations. The MAOA gene is located on the X chromosome—of which males possess only one copy—and it is challenging to determine which MAOA copy is activated in females; furthermore, COMT transcription is inhibited by estrogen. Therefore, studies that do not control for gender and heterozygosity in females are likely to be limited in their findings. The German study that identified an interaction related to schizotypy scores excluded heterozygotes, enhancing sensitivity to female participants. In contrast, the English study, which found no association, was conducted exclusively on males, evaluating the risk of schizophrenia, suggesting that the COMT × MAOA interaction may not be correlated with risk but rather with schizotypy alone, or it may not be significant in males. It remains unclear whether the Indian study specifically tested this interaction, which could indicate a false negative. Further research on this topic could provide valuable insights.

## 4. Discussion

Schizophrenia is a highly heterogeneous disease, and dopamine is still considered the most important neurotransmitter involved in the development of the disease. There is reason to suspect that different dopaminergic imbalances may be linked to certain disease phenotypes or symptoms, in addition to disease vulnerability. There is evidence suggesting potential differences influenced by gender. Moreover, these patterns are believed to interact with other neurotransmitter genes (i.e., 5HTR2A), indicating that schizophrenia is unlikely to solely result from dopamine dysregulation in all cases. Even though very few combinations have been investigated in more than one study, most observed associations (Figure 6) consist of a dopamine receptor and either a metabolizing enzyme or a transporter.

The most extensively investigated gene remains COMT, particularly the *Val158Met* polymorphism, which is functionally represented by different rates of frontal dopamine degradation. The risk allele G (*Val*), in conjunction with the DRD2 *Taq1A* A1 allele, correlates with reduced autoreceptor function and potentially higher striatal dopamine concentrations, yet high dopamine turnover in the frontal brain region, and represents a risk pattern. Notably, this combination exhibits a unique protective effect in Russian populations, warranting further investigation.

Another seemingly paradoxical effect arises from its interaction with DRD2 rs6275, where the *Met*/*Met* genotype is protective in conjunction with rs6275 CC and CT genotypes but poses a fourfold risk in the presence of the risk TT genotype (associated with affected DRD2 transcription [113]). In the presence of T allele(s) of DRD2 rs2283265, which result in an alternative DRD2 splicing and lower presynaptic short isoform expression [114], *Val* may contribute to developing the hypofrontality associated with SZ. The *Val*/*Val* genotype may also confer a degree of vulnerability in carriers of the T allele in the TH gene rs10770141 (which is associated with 30–40% higher TH expression), as a higher quantity of DA metabolized at a higher rate may potentially result in a build-up of neurotoxic waste-products. There is some evidence that both *Val*/*Val* and *Met*/*Met* may be associated with high levels of schizotypy and “unusual experiences”, depending on the variant of MAOA uVNTR, both being enzymes responsible for dopamine degradation. The combination of either high-activity (*Val*/*Val*, four-repeat) or low-activity (*Met*/*Met*, three-repeat) variants may both lead to developing schizotypal traits, yet these are functionally completely the opposite, being associated with lower (respectively, higher) overall available DA.

The above findings suggest that the relevance of the *Val158Met* polymorphism (rs4680) lies in the complex interactivity of the dopaminergic system. As per our review, the higher prefrontal dopamine turnover of the G (*Val*) allele is associated with disease vulnerability in conjunction with either decreased DRD2 autoreceptor function (*Taq1A* A1, rs2283265 T allele), higher TH expression (*C*-*824T*, T allele), higher DRD1 expression (rs11749676 G/GG), or high MAOA activity (uVNTR 4-repeat). Similarly, *Met* may be a risk factor in the presence of increased DRD2 autoreceptor function (*Taq1A* A2), dysfunctional DRD2 transcription (rs6277 TT), or low MAOA activity (uVNTR three-repeat). Often, heterozygosity appears at least mildly protective.

The interaction between Val158Met and Taq1A is particularly noteworthy, as numerous studies have explored their combined effects on various cognitive processes, including attention, memory, and decision-making thresholds—key elements in the development of psychotic experiences. Our review highlights two specific risk patterns (Val/Val × A1+ and Met/Met × A2/A2), both of which are associated with a tendency to be distracted by novel stimuli yet exhibit greater accuracy in processing distracting information [115]. In contrast, individuals with non-risk patterns demonstrate a quicker but more error-prone response to new stimuli.

Additionally, the Val/Val A1+ and Met/Met A2/A2 patterns appear to interact in behavioral timing, leading to earlier start times on tasks when larger rewards are at stake [116]. Both risk variants tend to have poorer working memory performance compared to the Val/Met × A1+ combination [87]. Interestingly, while the Met/Met A2/A2 pattern is linked to strong memory manipulation skills, this advantage diminishes in tasks that require cognitive flexibility [117].

Though interpreting these findings on cognitive processes should be conducted cautiously, as they would rely on some degree of speculative extrapolation, these risk patterns influence certain cognitive functions in similar ways. The link between these risk patterns and greater accuracy in processing novel auditory or visual stimuli might appear counterintuitive; however, it is possible that frequent positive reinforcement for correct interpretation of uncommon perceptions could foster overconfidence, even when these interpretations are incorrect. This tendency could, in turn, be loosely associated with the development of strong convictions in false beliefs or perceptual anomalies. Moreover, these patterns are associated with a heightened focus on novel and unexpected stimuli, quicker responses in high-reward situations, and relatively poorer working memory performance, all of which are cognitive traits that may contribute to the development of psychosis. For instance, they could translate to misinterpretations of environmental cues under high emotional stress, with a hindered ability to distinguish between relevant and irrelevant information, potentially fostering the cognitive distortions seen in psychotic episodes. More research into the exact mechanisms of interaction between cognitive processes and the development of psychosis would be of relevance for these two genetic patterns.

A complex dopaminergic gene pattern COMT × DRD2 × DDC × DAT suggests that dopaminergic epistasis might be related not only to disease risk, but perhaps more specifically to its delusional aspect.

There are insufficient data on specific SNPs to draw clear interpretations, particularly as the risk alleles were not stated. While no relevant studies have focused on the interactions between these specific SNPs, it is known that COMT and DRD2 significantly interact in cognitive processes. Furthermore, COMT and DAT also seem to interact in modulating working memory [118] and cognitive function [119], as well as in influencing cortisol response magnitude to stress and stress recovery [120]. Additionally, DRD2 and DAT interact to regulate prefrontal activity and affect the grey matter volume of the striatum [121]. In vivo, the interaction between the proteins they encode may contribute to dopamine neurotoxicity [122], and these gene interactions could be influenced by variations in dopamine synthesis, as indicated by DDC.

There appear to be numerous connections between these genes and cognitive functions, particularly in working memory and prefrontal activity, as well as in overreactivity to stress and neurotoxicity. This suggests that certain cognitive disturbances may contribute to delusional symptomatology, possibly stemming from inadequate cognitive responses to stress, as well as dopamine neurotoxicity. While altered levels of dopamine synthesis may influence this relationship, the specific mechanisms remain unclear. The three other genes within this complex are associated with serotonin, glutamate, and nitric oxide, indicating that, although this is a predominantly DA-related association, there may also be interactions with other neurotransmitter systems.

Another potential variant of interest might be DRD3 *Ser9Gly*, where the *Gly* is linked to increased dopamine binding affinity and is rather inconsistently associated with SZ risk. One of the studies under review also observed no independent association between this variant and SZ, yet it appears to have a significant interactive effect in carriers of uVNTR nine-repeat of the SLC6A3 (DAT1) gene. In the presence of reduced DAT expression (linked to the shorter nine-repeat variant), the *Ser9Gly* homozygotes have striking differences: *Ser*/*Ser* have a strongly reduced SZ risk, while *Gly*/*Gly* have a much higher vulnerability. It is generally considered that the *Gly* results in susceptibility to hyperdopaminergic response to negative experiences [123]. However, in conjunction with reduced transporter function and implicitly reduced DA reuptake, we hypothesize that the increased dopaminergic tone (which notably spares the prefrontal cortex, where both genes have reduced expression [124,125]) might predispose patients to developing SZ, particularly during periods of acute or prolonged stress. *Ser9Gly* appears to also interact with the functionally uncertain rs12516948 as well as potentially the DRD4 VNTR in Exon 1; however, the latter interaction might be wiser to be investigated in a larger serotonin–dopamine context.

While a notable amount of variant interactions cannot be functionally interpreted, we do notice some patterns, such as most interactions implicating a *receptor dysfunction* (particularly DRD2, followed by DRD3), two different classes of proteins (notably *receptor*-*metabolism* centered around DRD2–COMT, followed by *receptor*-*transporter* mainly consisting of DRD3–SLC6A3). Even though COMT × SLC6A3 (DAT1) has only been investigated by two different studies, the discovery of multiple interactive variants (most of which are functionally uncertain) implies the need for further research.

The recurring functional patterns are combinations with concurrent hyperdopaminergic effects in limbic/striatally expressed genes, or high striatal DA tone and low prefrontal dopamine. Due to the relatively small number of studies, no clear rule can be extracted.

Although there are currently insufficient data to make definitive conclusions, exploring these combinations could lead to the identification of future diagnostic markers and inform treatment strategies. Notably, some of the risk patterns identified are linked to dopamine imbalances—either too low or too high—while only a subset is associated with hyperdopaminergic activity. Given that most commonly used antipsychotics primarily reduce dopamine levels, this may contribute to the chronicity of the disorder and the treatment resistance many patients experience, thus impacting the choice between dopamine antagonists and partial agonists.

Understanding various risk patterns may be crucial for disease prevention and formulating new treatment strategies. For instance, individuals with the Val/Val genotype (rs4680) who also possess additional risk factors—such as the Taq1A A1/A1 genotype, genotypes associated with higher TH expression, etc.—could benefit from the inclusion of a COMT inhibitor in their treatment plans. This is particularly important, as Val/Val patients often exhibit poorer responses to antipsychotic medications [126,127] and experience more cognitive side effects [128]. While COMT inhibitors do reverse the cognitive differences between the two genotypes [129], they have not yet been investigated in SZ treatment in conjunction with the Val158Met genotype. Reversely, the combination of Met/Met and potential genetic risk factors (such Taq1A A2/A2 or dysfunctional DRD2 transcription) would indicate conventional antipsychotic treatment.

To further address the COMT/DRD2 risk patterns, research towards implementing screening tests comprised of working memory testing, auditory–visual distraction, and decision-making tasks could could be useful for identifying at-risk populations and initiating preventive strategies. An example of a possible early intervention would be developing targeted cognitive therapies that focus on enhancing working memory and error-tolerability training, as well as resilience to stress.

Furthermore, carriers of both the short SLC6A3 (DAT1) variant and the Ser9Gly Gly/Gly genotype may need to adopt a more proactive approach to manage stress and maintain proper mental hygiene. In Gly/Gly individuals, substances that act as dopamine reuptake inhibitors (DRIs) should be avoided—such as methylphenidate—and substance abuse disorders must be imperatively addressed, as these compounds can effectively mimic a potential risk genotype. It would be intriguing to investigate whether there is a link between the Gly/Gly genotype and the onset of substance-induced psychosis or its progression to schizophrenia, particularly in the context of DRI abuse.

Conversely, Gly/Gly patients show good response to antidopaminergic treatments. While the hyperdopaminergic Gly allele is associated with worse symptomatology (such as executive dysfunction and disorganization), it does respond better to antipsychotic treatment [130]. However, the difference does not hold true for aripiprazole, a partial agonist [131] or clozapine—antipsychotic with minimal DA binding [132]—which is noteworthy. Although no studies have yet explored the treatment response in patients with both the Gly/Gly genotype and the short (nine-repeat) DAT uVNTR variant, future research could examine the effectiveness of antipsychotics with strong antidopaminergic effects in comparison to partial dopamine agonists or those with minimal dopamine binding, such as clozapine.

As genetic testing continues to advance, becoming more affordable and accessible, we strongly advocate for further research into these genetic interactions. With enough accumulated data, meta-analyses could be conducted, enabling the consideration of these markers for use in screening. Additionally, a deeper understanding of their functional implications could pave the way for the development of new, targeted treatment strategies.

Earlier studies, including various meta-analyses, have often focused on single-gene associations with schizophrenia, leading to inconsistent and sometimes contradictory results. There is evidence to suggest that this inconsistency may be explained by epistatic interactions, suggesting that future research concerning genetic vulnerability to schizophrenia may be better approached from an interaction-focused standpoint. Furthermore, expanding the scope of genetic studies to include other neurotransmitter-related genes could provide a broader understanding of how multiple neurobiological pathways contribute to the disorder. Additionally, gene–gene interactions could be explored not only in relation to schizophrenia susceptibility but also in terms of treatment responses and side effects, which could help identify gene complexes that predict therapeutic outcomes or adverse reactions, allowing for more personalized approaches to treatment and prevention.

While current GWAS studies have insignificant results concerning dopamine-related genes, we believe the dopamine theory of schizophrenia should not yet be put to rest. Future large-scale studies may benefit from investigating dopaminergic interactions, while taking into account sex, neuropsychiatric comorbidities, and PANSS subscores in addition to the presence of a schizophrenia diagnosis. Seeking to uncover unique associations between genetic and phenotypic profiles, rather than a simplistic risk analysis, could be a more suitable approach to neuropsychiatric-related GWAS studies in general, as has been observed in the case of bipolar disorder in a GWAS association analysis [133].

## 5. Conclusions

This review expands on the dopaminergic theory of schizophrenia, suggesting the existence of multiple genetic patterns of vulnerability and highlighting the need for additional targeted research. In many cases, the protective status of certain individual alleles or genotypes is preserved in epistatic effects; however, this is not a rule. Some specific combinations that consist of generally considered “protective” alleles have been found to be associated with a higher risk instead, which could potentially explain the relatively inconsistent results in previous genetic studies. Simply investigating individual alleles may not be sufficient, due to the heterozygous effect, which has also been observed in some of the studies investigated. The review also suggests the need for an epistatic approach to genetic vulnerability in schizophrenia, as certain genes may not exhibit associations independently, but only in interaction with others. Moreover, conducting sex-dependent analyses appears to be of relevance in this domain.

Regarding future association studies, this review would propose investigating interactions between DRD2, DRD3, COMT, and SLC6A3 (DAT1), in particular the rs4680 (Val158Met), rs1800497 (Taq1A), rs6275 (His313His), rs6280 (Ser9Gly), rs6347, and DAT uVNTR polymorphisms. Elucidation of specific allele functionality, and consideration of interactions with other neurotransmitters, is also warranted. To further narrow down patterns, future studies could consider examining the genetic associations with specific symptom clusters and disease subtypes also. It is very likely that some genetic vulnerability patterns are only triggered in the presence of certain environmental factors (such as chronic stress, drug abuse [134], etc.), so controlling for non-genetic risk factors might also be necessary.

Schizophrenia may be better viewed not as a monolithic disease but rather as a collection of different phenotypes. Unique responses to different treatment and prevention approaches may be predicated on the effects of these phenotypes.

## Figures and Tables

**Figure 1 brainsci-14-01089-f001:**
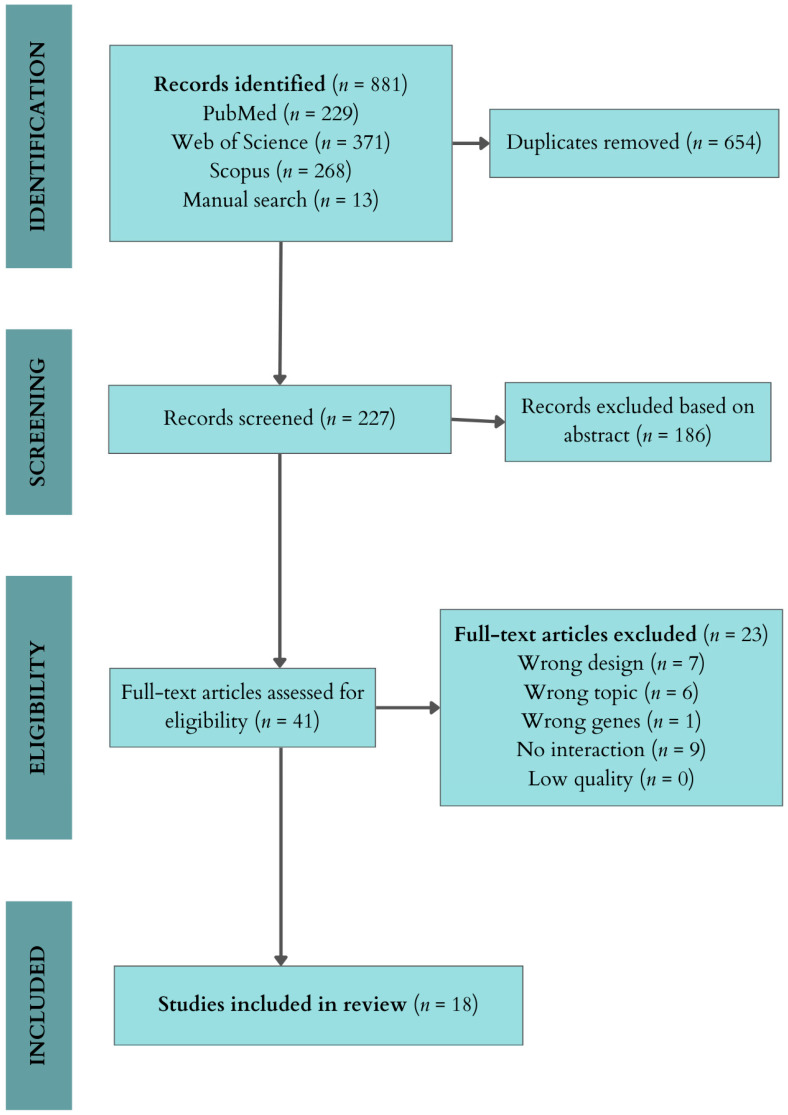
Flowchart of the study identification process.

**Figure 2 brainsci-14-01089-f002:**
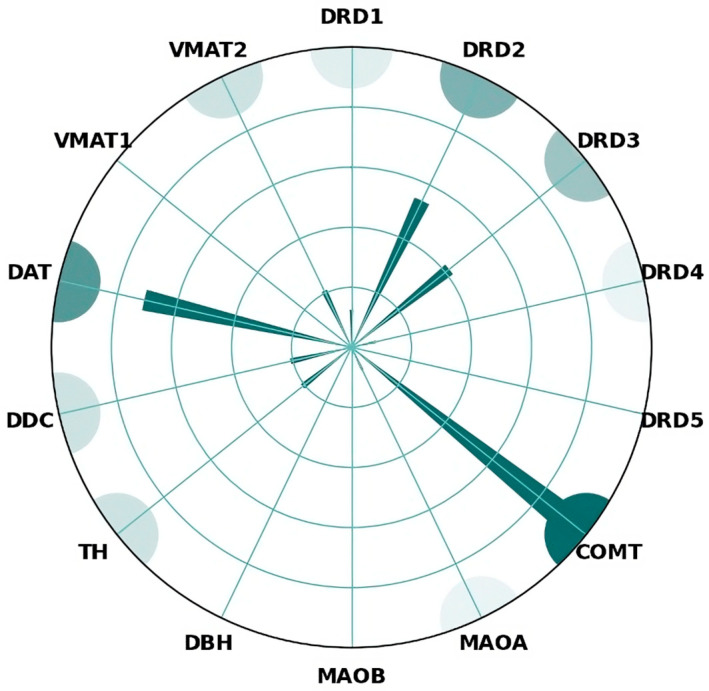
Polar bar chart of dopaminergic genes identified to participate in epistatic interactions. COMT, DAT, DRD2, and DRD3 emerged as the most influential in these interactive models. In contrast, no studies have identified significant epistatic associations involving DRD5, MAOB, DBH, or VMAT1. An interaction score is visually represented for each gene, where trend-level associations are given a lower weight but are still included in the overall aggregation.

**Figure 3 brainsci-14-01089-f003:**
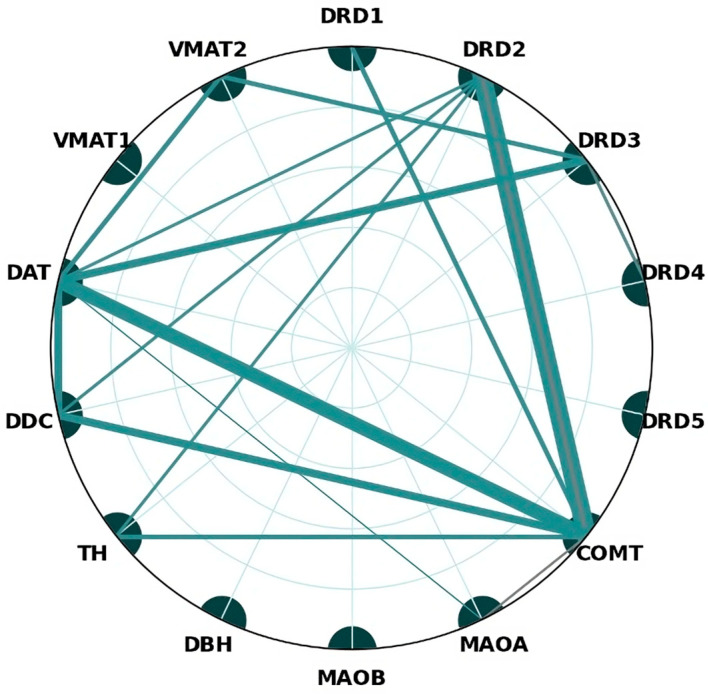
The observed genetic interactions within the dopaminergic system associated with schizophrenia risk are illustrated; the strength of these interactions is visually denoted by line weight, where trend-level associations and findings from lower-quality studies are given less emphasis, as indicated by thinner lines. The interaction between COMT and DRD2 is the most frequently reported finding among the included studies. These are aggregated scores considering all SNPs associated with a gene.

**Figure 4 brainsci-14-01089-f004:**
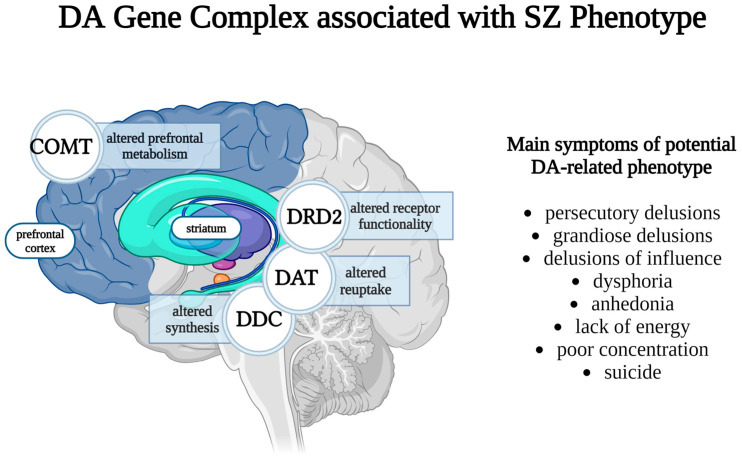
While the functional mechanisms are unclear, a DA gene complex may be associated with certain SZ symptoms, hinting at the idea that DA-associated SZ may represent one out of many different phenotypes. The complex consists of COMT (mainly expressed in the prefrontal cortex), DRD2 (mainly expressed in the striatum, olfactory tubercle, and nucleus accumbens), DAT (substantia nigra, ventral tegmental area, striatum, and nucleus accumbens), and DDC (hypothalamus, striatum, and locus coeruleus) [112].

**Figure 5 brainsci-14-01089-f005:**
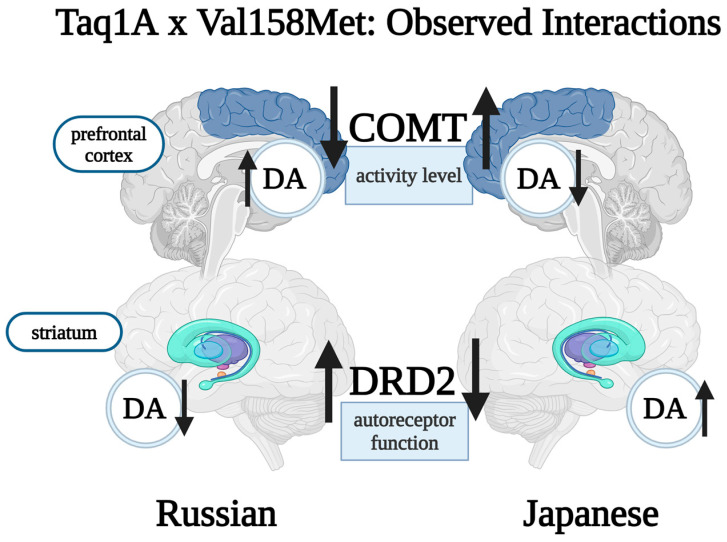
This review highlights two potential interactive effects of the COMTxDRD2 gene combination. The Japanese population (illustrated on the right) appears vulnerable to high COMT activity in the prefrontal brain (associated with the Val/Val genotype), combined with lower DRD2 receptor availability (associated with the A1 variant). This is functionally translated by lower prefrontal dopamine levels and higher striatal DA activity. The opposite effect may appear in Russians, however (left side), where lower COMT activity and higher DRD2 autoreceptor function led to higher prefrontal DA and lower striatal DA levels, and appear to represent a risk pattern [112].

**Figure 6 brainsci-14-01089-f006:**
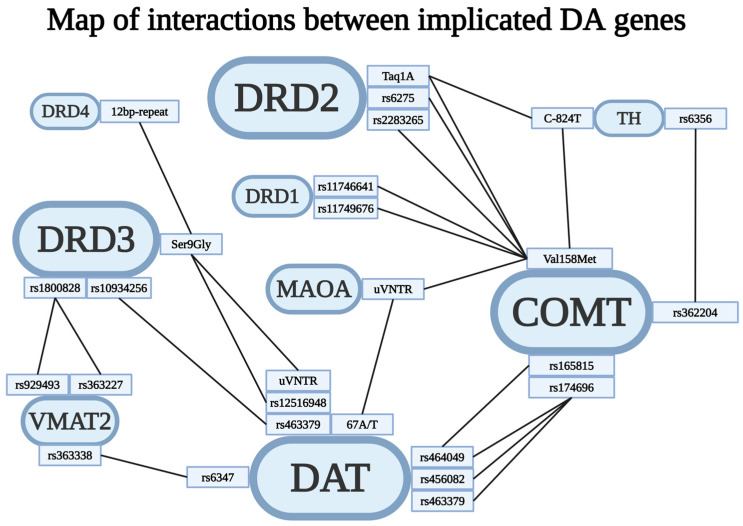
The graph depicts all of the observed interactions between pairs of gene variants within the dopaminergic system. Each gene’s relevant SNPs are shown, with lines representing the interactions between specific SNP pairs that have been discussed in this review. It is important to note that the COMT × DRD2 × DAT × DDC complex is not included in this illustration, see Figure 4 [112].

**Table 1 brainsci-14-01089-t001:** Characteristics of studies [94,95,96,97,98,99,100,101,102,103,104,105,106,107,108,109,110,111].

Title	Author	SNPs Investigated	Method Used	No. Participants	Risk of Bias (NOS Score ^1^)	Results
1. A network of dopaminergic gene variations implicated as risk factors for schizophrenia	Talkowski [98], 2008	DRD3: rs10934256, rs1800828, rs324030, rs6280, rs7625282, rs2134655; COMT: rs165815, rs174696; DAT: rs2078247, rs3756450, rs403636, rs463379, rs456082, rs464049, rs37022, rs6347, rs12516948; VMAT2: rs363338, rs363343, rs929493, rs4752045, rs363227	Logistic Regression: unconditional logit model (case-control), conditional logit model (family trios)	2956: 478 patients; 501 controls; 679 family trios	7 S **** C- E ***	Multiple associations between COMT × DAT, DAT × DRD3, DAT × VMAT2, DRD3 × VMAT2
2. Association study of schizophrenia and DRD3, DRD4, and HTR2A gene polymorphisms	Hwang [99], 2004	DRD3: rs6280; DRD4: 12-bp VNTR	Logistic Linear Regression	387: 145 patients (75 M, 70 F); 242 controls (126 M, 116 F)	8 S *** C ** E ***	DRD3 rs6280 × DRD4 12 bp VNTR
3. Association study of an SNP combination pattern in the dopaminergic pathway in paranoid schizophrenia: a novel strategy for complex disorders	Xu [100], 2004	COMT: rs165599, rs165656, rs165688, rs174682, rs174694, rs4818, rs740603, rs933269; MAOA: rs1801291, rs6323; MAOB: rs1040399; DBH: rs2005663, rs2007153; DDC: rs1451373, rs1451374, rs6263, rs921450	PESCP (potential effective SNP combination pattern), PEDE (potential effective dynamic effects)	476: 83 patients (34 M, 49 F); 108 controls; 95 family trios	7 S ** C ** E ***	No relevant associations
4. Common polymorphisms in dopamine-related genes combine to produce a ’schizophrenia-like’ prefrontal hypoactivity	Vercammen [94], 2014	DRD2: rs2283265; COMT: rs4680	Correlation (oligogenic score)	70: 27 patients (19 M, 8 F); 43 controls (19 M, 24 F)	6 S *** C- E ***	COMT rs4680 × DRD2 rs2283265
5. Dopaminergic foundations of schizotypy as measured by the German version of the Oxford-Liverpool Inventory of Feelings and Experiences (O-LIFE)—a suitable endophenotype of schizophrenia	Grant [97], 2013	COMT: rs4680; MAOA: uVNTR	GLM Analysis (general linear model)	288: 91 M, 197 F	Correlational (NOS not applicable)	COMT rs4680 × MAOA uVNTR (trend-level)
6. Dopaminergic pathway gene polymorphisms and genetic susceptibility to schizophrenia among north Indians	Srivastava [95], 2010	DRD1: rs4532, rs5330, rs5331, rs13306309, rs686; DRD2: rs1799732, rs1079597, rs1801028, rs2234689, rs1800497; DRD4: rs4646984, rs1800955, 48-bp VNTR; COMT: rs2075507, rs4633, rs4818, rs4680, rs362204; MAOA: 30-bp VNTR; MAOB: rs1799836; DBH: rs1611115, rs1108580, rs5320, rs4531, rs129882; TH: rs6356, rs28934579; DDC: rs3829897, rs6593010, rs11575542, rs11575553	Backwards Binary Logistic Regression Analysis	430: 215 patients (112 M, 103 F); 215 controls (130 M, 86 F)	6 S *** C- E ***	COMT rs362204 × TH rs6356
7. Genetic polymorphisms in the dopamine-2 receptor (DRD2), dopamine-3 receptor (DRD3), and dopamine transporter (SLC6A3) genes in schizophrenia: Data from an association study	Saiz [101], 2010	DRD2: rs1799732; DRD3: rs6280; DAT: uVNTR	Logistic Regression (multivariate), Wald statistic for significance	707: 286 patients (172 M, 114 F); 421 controls (216 M, 205 F)	7 S *** C- E ***	DRD3 rs6280 × DAT uVNTR
8. Genetic risks of schizophrenia identified in a matched case-control study	Oishi [102], 2021	DRD2: rs1800497; COMT: rs4680; TH: rs10770141	Multiple Logistic Regression	2544: 1272 patients (574 M, 698 F); 1272 controls (603 M, 669 F)	8 S *** C ** E ***	COMT rs4680 × DRD2 rs1800497, DRD2 rs1800497 × TH rs10770141
9. Genetic susceptibility to schizophrenia: role of dopaminergic pathway gene polymorphisms	Gupta [103], 2009	DRD2: rs1799732, rs4274224, rs12808482, rs11608185, rs2075652, rs1801028, rs6275, rs6277; COMT: rs3788319, rs737865, rs6269, rs4633, rs4818, rs4680, rs165599	MDR (multifactor dimensionality reduction)	422: 243 patients (147 M, 96 F); 179 controls (115 M, 64 F)	7 S ** C ** E ***	DRD2 rs6275 × COMT rs4680; DRD2 rs6275 × DRD2 rs4274224 × COMT rs4680 × COMT rs3788319 (trend-level)
10. Interaction of multiple gene variants and their effects on schizophrenia phenotypes	Cheah [104], 2016	DRD2: rs1800499, rs2734839, rs6277; DRD3: rs1800828, rs324035; COMT: rs165774, rs4646316, rs4680; DDC: rs1966839, rs2329371; DAT: rs11133767, rs40184, rs4975646, rs6347	Binary Logistic Regression	460: 235 patients (165 M, 70 F); 225 controls (104 M, 121 F)	8 S **** C * E ***	DRD2 rs2734839 × COMT rs4680 × DAT rs6347 × DDC rs1966839 (delusional symptom cluster); DDC rs2329371 × DDC rs1966839 × DAT rs11133767 × COMT rs4646316
11. Lack of association between schizophrenia and polymorphisms in dopamine metabolism and transport genes	Alvarez [105], 2010	MAOA: rs6323, 30-bp VNTR; DAT: rs2975226, 40-bp VNTR	MDR (multifactor dimensionality reduction)	532: 242 patients (132 M, 110 F); 290 controls (111 M, 179 F)	7 S *** C * E ***	DAT rs2975226 × MAOA uVNTR (trend-level)
12. No association between SLC6A2, SLC6A3, DRD2 polymorphisms and schizophrenia in the Han Chinese population	Bi [96], 2017	DRD2: rs2234689, rs7131056; DAT: rs3863145, rs2550956	MDR (multifactor dimensionality reduction), Line Regression	2068: 1034 patients (588 M, 446 F); 1034 controls (625 M, 409 F)	5 S ** C- E ***	No association
13. Polymorphisms of dopamine D2-like (D2, D3, and D4) receptors in schizophrenia	Ohara [106], 1996	DRD2: rs1801028; DRD3: rs6280; DRD4: 12-bp VNTR, 48-bp VNTR	Chi-square/Fisher’s Exact Test	274: 153 patients (77 M, 76 F); 121 controls (51 M, 70 F)	8 S **** C * E ***	No association
14. Potential genetic variants in schizophrenia: a Bayesian analysis	Hall [107], 2007	DRD2: rs1801028, rs1800496; DRD3: rs6280; MAOA: rs1800466; DBH: SNP000007898	Bayesian statistical models	192: 103 patients (64 M, 39 F); 89 controls (60 M, 29 F)	8 S **** C * E ***	No relevant associations
15. Preliminary evidence for association between schizophrenia and polymorphisms in the regulatory regions of the ADRA2A, DRD3 and SNAP-25 genes	Lochman [108], 2013	DRD1: rs4532, rs265981; DRD3: rs6280; DBH: rs2519152	GENECOUNTING (Likelihood-ratio test)	405: 192 patients (192 M); 213 controls (213 M)	7 S *** C * E ***	No relevant associations
16. Schizophrenia and functional polymorphisms in the MAOA and COMT genes: No evidence for association or epistasis	Norton [109], 2002	COMT: rs4680, rs2075507; MAOA: 30-bp VNTR, rs6323	Chi-square	486: 248 patients (248 M); 238 controls (238 M)	8 S *** C ** E ***	No association
17. Sexually dimorphic interaction between the DRD1 and COMT genes in schizophrenia	Hoenicka [110], 2010	DRD1: rs11746641, rs11749676, rs251937, rs12518222, rs4867798; COMT: rs4680	Stepwise Logistic Regression	701: 337 patients (226 M, 111 F); 364 controls (171 M, 193 F)	6 S *** C- E ***	COMT rs4680 × DRD1 rs11746641, COMT rs4680 × DRD1 rs11749676
18. The role of intergenic interactions of neurotrophic and neurotransmitter system genes in the development of susceptibility to paranoid schizophrenia	Gareeva [111], 2020	DRD2: rs1800497, rs6275; DRD3: rs6280; DRD4: rs747302, 12-bp VNTR; COMT: rs4680, rs4818	MDR (multifactor dimensionality reduction)	606: 257 patients (137 M, 120 F); 349 controls	7 S ** C ** E ***	COMT rs4680 × DRD2 rs1800497

^1^ The Newcastle–Ottawa scale scores have been summarized by selection quality (S, rated out of a maximum of four stars), comparability quality (C, rated out of a maximum of two stars), and exposure quality (E, rated out of a maximum of three stars).

**Table 2 brainsci-14-01089-t002:** Genes investigated and significant SNPs.

Gene	Interaction	Gene Variant
DRD1	COMT	rs11749676, rs11746641
DRD2	COMT, DDC, DAT, TH	rs1800497, rs6275, rs2734839, rs2283265
DRD3	DRD4, DAT, VMAT2	rs6280, rs1800828, rs10934256
DRD4	DRD3	12 bp VNTR (D4E1)
DRD5	-	-
COMT	DRD1, DRD2, MAOA, TH, DDC, DAT	rs4680, rs362204, rs4646316, rs165815, rs174696
MAOA	COMT, DAT	30 bp VNTR (promoter)
MAOB	-	-
DBH	-	-
TH	DRD2, COMT	rs10770141, rs6356
DDC	DRD2, COMT, DAT	rs1966839, rs2329371
DAT	DRD2, DRD3, COMT, MAOA, DDC, VMAT2	rs6347, 40 bp VNTR, rs11133767, rs463379, rs12516948, rs456082, rs464049, rs456082
VMAT1	-	-
VMAT2	DRD3, DAT	rs363227, rs363338, rs929493

## Data Availability

Data sharing is not applicable.

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
