# Peer review of "Dopaminergic Epistases in Schizophrenia"

_brainsci, 2024, doi:10.3390/brainsci14111089_

Round 1

Reviewer 1 Report

Comments and Suggestions for Authors

One important concern is that it is a systematic review without a PROSPERO registration. It reinforces to the journal and to the reader the originality and relevance of the study. Considering that registration on PROSPERO is free of charges and has been strongly stimulated by the journals, why did not the authors register it? Could you give to the readers an idea about where this study is regarding reviews about this topic?

The introduction could be improved. The authors gave a good basis for the choice of each gene. However, another main point in this work is the investigation of the epistatic effects on schizophrenia and the way the authors addressed it in the introduction can be improved.  In this sense, in the abstract and in the introduction is not clear why the gene-gene interaction should be investigated in schizophrenia. In the introduction the authors named several genes and polymorphisms which were already associated to schizophrenia but did not provide evidence about the gene-gene interaction among them and/or schizophrenia or the relationship with epistatic effects and neuropsychiatric disorders for example. The readers would be benefited by a more detailed introduction about epistasis and its relationship with schizophrenia.

As a suggestion, the table 1 could be improved with the NOS calculated for each study and with the main results regarding the epistatic effects for each one.

Author Response

Comment 1: One important concern is that it is a systematic review without a PROSPERO registration. It reinforces to the journal and to the reader the originality and relevance of the study. Considering that registration on PROSPERO is free of charges and has been strongly stimulated by the journals, why did not the authors register it? Could you give to the readers an idea about where this study is regarding reviews about this topic?

Response 1: We recognize the importance of PROSPERO registration in enhancing the transparency and credibility of systematic reviews and are committed to ensuring that future studies comply with this standard. Unfortunately, at the onset of this review, our team was not fully aware of the increasing emphasis on PROSPERO registration for systematic reviews, which has become a common practice in recent years. See page 2, lines 63-65: 

“To the best of our knowledge, no other reviews have specifically examined the interactions between dopamine-related genes and schizophrenia, nor are there any broader reviews that address this particular research question.”

Comment 2: The introduction could be improved. The authors gave a good basis for the choice of each gene. However, another main point in this work is the investigation of the epistatic effects on schizophrenia and the way the authors addressed it in the introduction can be improved.  In this sense, in the abstract and in the introduction is not clear why the gene-gene interaction should be investigated in schizophrenia. In the introduction the authors named several genes and polymorphisms which were already associated to schizophrenia but did not provide evidence about the gene-gene interaction among them and/or schizophrenia or the relationship with epistatic effects and neuropsychiatric disorders for example. The readers would be benefited by a more detailed introduction about epistasis and its relationship with schizophrenia.

Response 2: Thank you for your valuable feedback! We understand the importance of providing a clearer rationale for investigating gene-gene interactions in schizophrenia, and we have revised the introduction to address this. We believe these revisions will offer readers a better understanding of the significance of exploring epistatic effects in this context. See page 5, lines 211-239: 

“Schizophrenia is a highly multifactorial disorder, and while numerous studies have focused on identifying genetic risk factors through single-gene analyses, this approach is overly simplistic and has often produced conflicting results, suggesting it is unlikely to fully capture the complexity of the disease. However, a genetic basis for schizophrenia is well-established, as shown by evidence from twin and adoption studies [131], indicating that genetic vulnerability is likely polygenic [132].

Given the long-recognized association between schizophrenia and dopamine imbalances, we believe that exploring multigenic interactions among genes directly related to key components of the dopamine system (such as receptors, enzymes, and transporters) provides a valuable starting point for understanding the genetic basis of the disorder. This focus does not exclude the potential role of other genes or environmental factors, which should be further explored.

Research has shown that dopamine-related genes interact in ways that influence cognitive and behavioral processes relevant to schizophrenia. For example, interactions between DRD2 and COMT affect intra-network connectivity in the default-mode and salience networks [124], while DBH and MAOA influence attentional bias toward negative expressions [125]. Social cooperation has also been linked to interactions between DRD2 and COMT [126]. Additionally, DRD4 and COMT together impact prefrontal response control, even though each gene may not have a significant effect individually [127], and variations in DRD2 and COMT combinations are associated with differences in working memory performance [128].

There is also evidence of interactions among DRD2, DRD4, and COMT in the response to antipsychotic treatment [129]. For instance, the combination of the Met allele of COMT and the 120-bp allele of DRD4 is associated with better responses to clozapine, but only when these alleles are present together [130]. Therefore, investigating multigenic interactions is essential not only for identifying risk factors for schizophrenia but also for gaining a deeper understanding of the disease. In the context of dopamine-related gene interactions, such research could provide insights into the dopamine imbalances that underlie schizophrenia.”

Comment 3: As a suggestion, the table 1 could be improved with the NOS calculated for each study and with the main results regarding the epistatic effects for each one.

Response 3: Thank you for your suggestion. Summarizing the results of each study in a table is indeed valuable for enhancing clarity. Initially, we chose not to include a table due to concerns about formatting challenges, but we have now added one. We hope it is easily readable, and we anticipate that the printed version may present it in a readable format. See page 9, Table 1.

Reviewer 2 Report

Comments and Suggestions for Authors

Major Concerns:

  1. Comprehensiveness of Data Synthesis:

    • The systematic review covers a significant number of dopaminergic genes in relation to schizophrenia, but the integration of results from various studies lacks depth. While the article presents positive and negative associations, it does not delve deeply into the underlying causes of discrepancies between studies (e.g., differing methodologies, population differences, or environmental factors). A more rigorous analysis that addresses these variations would strengthen the review.
  2. Clarity on Epistasis Mechanisms:

    • The focus of the review is on epistatic interactions, but the mechanistic explanations of how gene-gene interactions lead to specific schizophrenia phenotypes are not sufficiently explored. For example, the impact of identified gene combinations (e.g., DRD2 and COMT) on dopamine signaling and their connection to schizophrenia subtypes requires more detailed discussion. Without this, the findings seem speculative rather than conclusive.
  3. Reproducibility of Methods:

    • The search strategy is described but lacks the necessary detail for full reproducibility. Specific dates for literature searches, exact Boolean operators, and any database filters used should be provided. This would ensure that future researchers can replicate the study selection process.
  4. Risk of Bias and Study Limitations:

    • While the review uses the Newcastle-Ottawa scale for assessing study quality, there is insufficient discussion of the biases in the included studies (e.g., publication bias, small sample sizes, ethnic diversity). Additionally, some studies mentioned as having "good quality" do not justify the criteria used to rank them. A more critical examination of these biases and limitations would give readers a better understanding of how they might affect the conclusions.
  5. Interpretation of Results:

    • The review highlights several gene interactions (e.g., COMT x DRD2) but does not clearly explain how these findings can be applied in clinical settings. For instance, how do these genetic interactions translate into potential treatment strategies or diagnostic markers for schizophrenia? A more direct connection between the findings and their potential clinical relevance would add significant value.

Minor Concerns:

  1. Terminology and Abbreviation Use:

    • Some terms and abbreviations (e.g., "SZ" for schizophrenia, "epistasis") are not introduced early enough or consistently used. This could confuse readers unfamiliar with specific genetic terms. Ensure that all abbreviations are defined clearly at first use and are used consistently throughout.
  2. Figures and Data Presentation:

    • Figures, such as those illustrating gene interactions (e.g., Figure 3), are crucial for understanding the epistatic relationships described in the review. However, these figures could benefit from improved resolution and clearer labeling of pathways and interactions. Additionally, the legends should provide more context for interpreting the figures, especially for complex genetic networks.
  3. Citations and References:

    • Some references appear out of order in the text, and there is inconsistent formatting of the citations. Ensure that the citation numbering aligns with the order of appearance in the text, and check that all references adhere to the journal’s formatting requirements.
  4. Discussion Depth:

    • The discussion on the relevance of gene-gene interactions is promising, but the review could delve further into how these findings compare with previous genetic research on schizophrenia. Additionally, more emphasis on future research directions and potential pitfalls in current genetic studies would enhance the paper’s depth.

Comments on the Quality of English Language

Need English editing.

Author Response

Comment 1: The systematic review covers a significant number of dopaminergic genes in relation to schizophrenia, but the integration of results from various studies lacks depth. While the article presents positive and negative associations, it does not delve deeply into the underlying causes of discrepancies between studies (e.g., differing methodologies, population differences, or environmental factors). A more rigorous analysis that addresses these variations would strengthen the review.

Response 1: Thank you very much for your observation, we hope we have now added adequate discussions on the discrepancies between the findings. See page 17, lines 470-479:
“These results are not replicated in the Tatar group [113], possibly due to different environmental and cultural factors, or competing genetic interactions.
No epistatic effect between Taq1A and Val158Met have been observed in Indian [101], and Australian [105] studies that have investigated multiple interactions. Taq1A Val/Met interactions may not have been investigated in Indian populations because the criteria for selecting interactions were not clearly stated; thus, it’s possible that only SNPs significant for single-gene testing were included for further interaction analysis. In Australian populations, the large number of genes analyzed under linear regression may have led to the exclusion of Taq1A Val/Met interactions, as any potential associations could have been overshadowed by larger combinations of genes (5 SNPs or more).“
Page 19, lines 561-566:
“No association was found between DRD3 Ser9Gly and the DRD4 12bp-repeat when analyzed individually in a Japanese cohort [108]. This absence of an association might be due to differences between Korean and Japanese populations; however, it is also possible that the interaction between DRD3 and DRD4 in relation to schizophrenia risk is influenced by the serotonergic system, making it less significant when considered in isolation.”
Page 19, lines 584-598:
“While the interaction between COMT Val158Met and MAOA uVNTR may be linked to schizotypy, this association has not been found to correlate with the development of schizophrenia in Indian [101] and English [111] populations. The MAOA gene is located on the X chromosome—of which males possess only one copy— and it is challenging to determine which MAOA copy is activated in females, furthermore, COMT transcription is inhibited by estrogen. Therefore, studies that do not control for gender and heterozygosity in females are likely to be limited in their findings. The German study that identified an interaction related to schizotypy scores excluded heterozygotes, enhancing sensitivity to female participants. In contrast, the English study, which found no association, was conducted exclusively on males, evaluating the risk of schizophrenia, suggesting that the COMT x MAOA interaction may not be correlated with risk but rather with schizotypy alone, or it may not be significant in males. It remains unclear whether the Indian study specifically tested this interaction, which could indicate a false negative. Further research on this topic could provide valuable insights.”

Comment 2: The focus of the review is on epistatic interactions, but the mechanistic explanations of how gene-gene interactions lead to specific schizophrenia phenotypes are not sufficiently explored. For example, the impact of identified gene combinations (e.g., DRD2 and COMT) on dopamine signaling and their connection to schizophrenia subtypes requires more detailed discussion. Without this, the findings seem speculative rather than conclusive.

Response 2: Thank you for your feedback. We have expanded the discussion to better highlight the connection between these interactions and specific phenotypes. See page 21, lines 652-682:
“The interaction between Val158Met and Taq1A is particularly noteworthy, as numerous studies have explored their combined effects on various cognitive processes, including attention, memory, and decision-making thresholds—key elements in the development of psychotic experiences. Our review highlights two specific risk patterns (Val/Val x A1+ and Met/Met x A2/A2), both of which are associated with a tendency to be distracted by novel stimuli yet exhibit greater accuracy in processing distracting information [133]. In contrast, individuals with non-risk patterns demonstrate a quicker but more error-prone response to new stimuli.
Additionally, the Val/Val A1+ and Met/Met A2/A2 patterns appear to interact in behavioral timing, leading to earlier start times on tasks when larger rewards are at stake [134]. Both risk variants tend to have poorer working memory performance compared to the Val/Met x A1+ combination [135]. Interestingly, while the Met/Met A2/A2 pattern is linked to strong memory manipulation skills, this advantage diminishes in tasks that require cognitive flexibility [136].
Though interpreting these findings on cognitive processes should be done cautiously, as they would rely on some degree of speculative extrapolation, it is clear that these risk patterns influence certain cognitive functions in similar ways. The link between these risk patterns and greater accuracy in processing novel auditory or visual stimuli might appear counterintuitive; however, it is possible that frequent positive reinforcement for correct interpretation of uncommon perceptions could foster overconfidence, even when these interpretations are incorrect. This tendency could, in turn, be loosely associated with the development of strong convictions in false beliefs or perceptual anomalies. Moreover, these patterns are associated with a heightened focus on novel and unexpected stimuli, quicker responses in high-reward situations, and relatively poorer working memory performance, all of which are cognitive traits that may contribute to the development of psychosis. For instance, they could translate to misinterpretations of environmental cues under high emotional stress, with a hindered ability to distinguish between relevant and irrelevant information, potentially fostering the cognitive distortions seen in psychotic episodes. More research into the exact mechanisms of interaction between cognitive processes and development of psychosis would be of particular relevance for these two genetic patterns.”
Page 21, lines 687-706:
“There is insufficient data on specific SNPs to draw clear interpretations, particularly as the risk alleles were not stated. While no relevant studies have focused on the interactions between these specific SNPs, it is known that COMT and DRD2 significantly interact in cognitive processes. Furthermore, COMT and DAT also seem to interact in modulating working memory [137] and cognitive function [138], as well as influencing cortisol response magnitude to stress and stress recovery [139]. Additionally, DRD2 and DAT interact to regulate prefrontal activity and affect the grey matter volume of the striatum [140]. In vivo, the interaction between the proteins they encode may contribute to dopamine neurotoxicity [141], and these gene interactions could be influenced by variations in dopamine synthesis, as indicated by DDC.
There appear to be numerous connections between these genes and cognitive functions, particularly in working memory and prefrontal activity, as well as in overreactivity to stress and neurotoxicity. This suggests that certain cognitive disturbances may contribute to delusional symptomatology, possibly stemming from inadequate cognitive responses to stress, as well as dopamine neurotoxicity. While altered levels of dopamine synthesis may influence this relationship, the specific mechanisms remain unclear. The three other genes within this complex are associated with serotonin, glutamate, and nitric oxide, indicating that, although this is a predominantly DA-related association, there may also be interactions with other neurotransmitter systems.”

Comment 3: The search strategy is described but lacks the necessary detail for full reproducibility. Specific dates for literature searches, exact Boolean operators, and any database filters used should be provided. This would ensure that future researchers can replicate the study selection process.

Response 3: Thank you for your feedback; indeed, after reviewing the section in question, we decided the specific literature searching method was not clear enough. See page 5, lines 273-308:
“273 individual searches have been performed until September 2023 in PubMed (5 sept.), Scopus (12 sept.) and Web of Science (5 sept.). The searches were composed of complex keywords for:  
1. each combination of 2 of the 14 genes under review (totaling 91 different complex keywords) connected through the operator “AND”; the complex keywords associated with each gene are as follows:  

(DRD1 OR “dopamine receptor D1” OR “dopamine D1 receptor”) 
(DRD2 OR “dopamine receptor D2” OR “dopamine D2 receptor”) 
(DRD3 OR “dopamine receptor D3” OR “dopamine D3 receptor”) 
(DRD4 OR “dopamine receptor D4” OR “dopamine D4 receptor”) 
(DRD5 OR “dopamine receptor D5” OR “dopamine D5 receptor”) 
(COMT OR Catechol-O-Methyltransferase) 
(“MAO-A” OR “monoamine oxidase A” OR “MAO A” OR MAOA) 
(“MAO-B” OR “monoamine oxidase B” OR “MAO B” OR MAOB) 
(DBH OR “Dopamine beta-hydroxylase” OR “dopamine beta-monooxygenase”) 
(TH OR “tyrosine hydroxylase” OR “tyrosine 3-monooxigenase”) 
(DDC OR “dopa decarboxylase” OR “tryptophan decarboxylase” OR “Aromatic L-amino acid decarboxylase” OR AADC OR AAAD OR “5-hydroxytryptophan decarboxylase”) 
(DAT OR “dopamine transporter” OR “dopamine active transporter” OR SLC6A3 OR “Solute Carrier Family 6 Member 3” OR “DA Transporter”) 
(VMAT1 OR “Vesicular monoamine transporter 1” OR “chromaffin granule amine transporter” OR CGAT OR “solute carrier family 18 member 1” OR SLC18A1 OR “VMAT 1” OR “VMAT-1”) 
(VMAT2 OR “Vesicular monoamine transporter 2” OR “solute carrier family 18 member 2” OR SLC18A2 OR “VMAT 2” OR “VMAT-2”) 
2. psychosis: “schizo* OR psychosis OR psychotic”,  
3. risk: “vulnerability OR predisposition OR susceptibility OR risk OR proneness”, and  
4. epistatic interaction: “epistasis OR “genexgene” OR “gene-gene” OR interact* OR epistatic OR GxG”, 
connected with the operator “AND”, without use of database filters, ex. (DRD1 OR “dopamine receptor D1” OR “dopamine D1 receptor”) AND (DRD2 OR “dopamine receptor D2” OR “dopamine D2 receptor”) AND (schizo* OR psychosis OR psychotic AND (vulnerability OR predisposition OR susceptibility OR risk OR proneness) AND (epistasis OR “genexgene” OR “gene-gene” OR interact* OR epistatic OR GxG).”

Comment 4: While the review uses the Newcastle-Ottawa scale for assessing study quality, there is insufficient discussion of the biases in the included studies (e.g., publication bias, small sample sizes, ethnic diversity). Additionally, some studies mentioned as having "good quality" do not justify the criteria used to rank them. A more critical examination of these biases and limitations would give readers a better understanding of how they might affect the conclusions.

Response 4: Thank you for your feedback. We have revised the manuscript to include a more detailed examination of the limitations in the included studies and we have clarified the criteria used to assess study quality with the Newcastle-Ottawa scale, providing more justification for how studies were ranked. See page 8, lines 325-347:
“For risk of bias assessment in case-control studies, we used the Newcastle-Ottawa scale (NOS) [95], which assesses quality based on three main categories: selection of study groups, comparability of groups, and ascertainment of exposures. Studies deemed "good quality" generally had robust sampling methods, clear criteria for genetic testing, and appropriate adjustment for confounding variables. Studies rated as "good quality" typically featured well-defined inclusion criteria and adequate control for confounders (such as age, gender, and socioeconomic status). Most studies were of good quality (13) with good selection, comparability and exposure overall; 3 were identified as high-risk of bias (poor comparability [99, 101, 107], with one study having relatively weak selection criteria [107]); none were considered subject to a very high risk of bias. One correlational study [100] could not be evaluated using the NOS scale, and its main limitation was the lack of a population representative of the general public, as the sample primarily consisted of academic staff and students, particularly from psychology. Additionally, schizotypy was assessed through a self-report scale, which is less reliable than DSM-based diagnoses. One study had a small sample size [99], which may lead to less reliable conclusions due to low statistical power, increasing the risk of false positives or false negatives. Additionally, most studies focused on a single population, meaning the findings might only be applicable to those specific groups, as different genetic backgrounds and environments could influence gene interactions. Due to the genetic genotyping methods used, all studies received the maximum score for exposure. It is also important to consider that studies with positive results are more likely to be published, which could lead to an overestimation of the effects reported in this review.”

Comment 5: The review highlights several gene interactions (e.g., COMT x DRD2) but does not clearly explain how these findings can be applied in clinical settings. For instance, how do these genetic interactions translate into potential treatment strategies or diagnostic markers for schizophrenia? A more direct connection between the findings and their potential clinical relevance would add significant value.
Response 5: Although these findings are not yet ready for direct clinical application, we have aimed to outline some of the research steps needed to bridge the gap between our current understanding of dopaminergic gene-gene interactions and their potential use in screening, identifying specific risk factors, and developing targeted treatment and early intervention strategies. See page 9, lines 735-784:
“Although there is currently insufficient data to make definitive conclusions, exploring these combinations could lead to the identification of future diagnostic markers and inform treatment strategies. Notably, some of the risk patterns identified are linked to dopamine imbalances—either too low or too high—while only a subset is associated with hyperdopaminergic activity. Given that most commonly used antipsychotics primarily reduce dopamine levels, this may contribute to the chronicity of the disorder and the treatment resistance many patients experience, thus impacting the choice between dopamine antagonists and partial agonists.
Understanding various risk patterns may be crucial for disease prevention and formulating new treatment strategies. For instance, individuals with the Val/Val genotype (rs4680) who also possess additional risk factors—such as the Taq1A A1/A1 genotype, genotypes associated with higher TH expression etc.—could benefit from the inclusion of a COMT inhibitor in their treatment plans. This is particularly important as Val/Val patients often exhibit poorer responses to antipsychotic medications [119,120] and experience more cognitive side effects [121]. While COMT inhibitors do reverse the cognitive differences between the two genotypes [142], they have not yet been investigated in SZ treatment in conjunction with the Val158Met genotype. Reversely, the combination of Met/Met and potential genetic risk factors (such Taq1A A2/A2 or dysfunctional DRD2 transcription) would indicate conventional antipsychotic treatment.
To further address the COMT/DRD2 risk patterns, research towards implementing screening tests comprised of working memory testing, auditory-visual distraction and decision-making tasks could could be useful for identifying at-risk populations and initiating preventive strategies. An example of a possible early intervention would be developing targeted cognitive therapies that focus on enhancing working memory and error-tolerability training, as well as resilience to stress.
Furthermore, carriers of both the short SLC6A3 (DAT1) variant and the Ser9Gly Gly/Gly genotype may need to adopt a more proactive approach to manage stress and maintain proper mental hygiene. In Gly/Gly individuals, substances that act as dopamine reuptake inhibitors (DRIs) should be avoided –such as methylphenidate– and substance abuse disorders must be imperatively addressed, as these compounds can effectively mimic a potential risk genotype. It would be intriguing to investigate whether there is a link between the Gly/Gly genotype and the onset of substance-induced psychosis or its progression to schizophrenia, particularly in the context of DRI abuse. 
Conversely, Gly/Gly patients show good response to antidopaminergic treatments. While the hyperdopaminergic Gly allele is associated with worse symptomatology (such as executive dysfunction and disorganization) it does respond better to antipsychotic treatment [143]. However, the difference does not hold true for aripiprazole, a partial agonist [144] or clozapine –antipsychotic with minimal DA binding– [145], which is noteworthy. Although no studies have yet explored the treatment response in patients with both the Gly/Gly genotype and the short (9-repeat) DAT uVNTR variant, future research could examine the effectiveness of antipsychotics with strong antidopaminergic effects, in comparison to partial dopamine agonists or those with minimal dopamine binding, such as clozapine. 
As genetic testing continues to advance, becoming more affordable and accessible, we strongly advocate for further research into these genetic interactions. With enough accumulated data, meta-analyses could be conducted, enabling the consideration of these markers for use in screening. Additionally, a deeper understanding of their functional implications could pave the way for the development of new, targeted treatment strategies.”

Comment 6: Some terms and abbreviations (e.g., "SZ" for schizophrenia, "epistasis") are not introduced early enough or consistently used. This could confuse readers unfamiliar with specific genetic terms. Ensure that all abbreviations are defined clearly at first use and are used consistently throughout.
Response 6: Thank you for your observation. We have attempted to add more clarity to the various terms used throughout. See page 2.

Comment 7: Figures, such as those illustrating gene interactions (e.g., Figure 3), are crucial for understanding the epistatic relationships described in the review. However, these figures could benefit from improved resolution and clearer labeling of pathways and interactions. Additionally, the legends should provide more context for interpreting the figures, especially for complex genetic networks.
Response 7: The figures have been updated with higher-resolution versions, and additional labels have been included to improve clarity. We also added more contextual information to the figures; however, succinctly conveying the complex interactions in a way that allows readers to fully grasp them without relying heavily on the text remains a challenge. See page 14, Figure 2:
“Polar bar chart of dopaminergic genes identified to participate in epistatic interactions. COMT, DAT, DRD2, and DRD3 emerged as the most influential in these interactive models. In contrast, no studies have identified significant epistatic associations involving DRD5, MAOB, DBH, or VMAT1. An interaction score is visually represented for each gene, where trend-level associations are given a lower weight but are still included in the overall aggregation.”
Page 14, Figure 3:
“The observed genetic interactions within the dopaminergic system associated with schizophrenia risk are illustrated; the strength of these interactions is visually denoted by lineweight, where trend-level associations and findings from lower-quality studies are given less emphasis, as indicated by thinner lines. The interaction between COMT and DRD2 is the most frequently reported finding among the included studies. These are aggregated scores considering all SNPs associated with a gene.”
Page 20, Figure 6:
“The graph depicts all observed interactions between pairs of gene variants within the dopaminergic system. Each gene’s relevant SNPs are shown, with lines representing the interactions between specific SNP pairs that have been discussed in this review. It is important to note that the COMT x DRD2 x DAT x DDC complex is not included in this illustration, see Figure 4 [123].”

Comment 8: Some references appear out of order in the text, and there is inconsistent formatting of the citations. Ensure that the citation numbering aligns with the order of appearance in the text, and check that all references adhere to the journal’s formatting requirements.

Response 8: Thank you for bringing this to our attention, and we apologize for the oversight. We have updated the citation format, but due to time constraints and the need for careful accuracy, we were unable to properly reorder the citations at this review phase.

Comment 9: The discussion on the relevance of gene-gene interactions is promising, but the review could delve further into how these findings compare with previous genetic research on schizophrenia. Additionally, more emphasis on future research directions and potential pitfalls in current genetic studies would enhance the paper’s depth.
Response 9: Thank you for your feedback. We have included a brief discussion comparing our findings with previous research and highlighted some promising general research directions. For more specific future research recommendations focused on genetic combinations of interest, please refer to Comment no. 5. See page 23, lines 785-805:
“Earlier studies, including various meta-analyses, have often focused on single-gene associations with schizophrenia, leading to inconsistent and sometimes contradictory results. There is evidence to suggest that his inconsistency may be explained by epistatic interactions, suggesting that future research concerning genetic vulnerability to schizophrenia may be better approached from an interaction-focused standpoint. Furthermore, expanding the scope of genetic studies to include other neurotransmitter-related genes could provide a broader understanding of how multiple neurobiological pathways contribute to the disorder. Additionally, gene-gene interactions could be explored not only in relation to schizophrenia susceptibility but also in terms of treatment responses and side effects, which could help identify gene complexes that predict therapeutic outcomes or adverse reactions, allowing for more personalized approaches to treatment and prevention. 
While current GWAS studies have insignificant results concerning dopamine-related genes, we believe the dopamine theory of schizophrenia should not yet be put to rest. Future large-scale studies may benefit from investigating dopaminergic interactions, while taking into account sex, neuropsychiatric comorbidities, and PANSS subscores in addition to the presence of a schizophrenia diagnosis. Seeking to uncover unique associations between genetic and phenotypic profiles, rather than a simplistic risk analysis could be a more suitable approach to neuropsychiatric-related GWAS studies in general, as has been observed in the case of bipolar disorder in a GWAS association analysis [146].”

Round 2

Reviewer 2 Report

Comments and Suggestions for Authors

My concerns have been well addressed.